# Synergistic Ag/g–C_3_N_4_ H_2_O_2_ System for Photocatalytic Degradation of Azo Dyes

**DOI:** 10.3390/molecules29163871

**Published:** 2024-08-15

**Authors:** Yajing Wang, Wen Yang, Kun Ding

**Affiliations:** 1College of Resources and Environment, University of Chinese Academy of Sciences, Beijing 101408, China; wangyajing18@mails.ucas.ac.cn; 2College of Environmental and Energy Engineering, Anhui Jianzhu University, Hefei 230601, China; yw99@stu.ahjzu.edu.cn

**Keywords:** graphitic carbon nitride, silver, photocatalysis, H_2_O_2_ system, azo dyes, dynamics

## Abstract

Graphitic carbon nitride (g-C_3_N_4_), known for being nontoxic, highly stable, and environmentally friendly, is extensively used in photocatalytic degradation technologies. Silver nanoparticles effectively capture the photogenerated electrons in g-C_3_N_4_, enhancing the photocatalytic efficiency. This study primarily focused on synthesizing graphitic carbon nitride via thermal polymerization and depositing noble metal silver onto g-C_3_N_4_ through photoreduction. Methyl orange (MO) and methylene blue (MB) were targeted as the pollutants in the photocatalytic experiments under visible light in conjunction with a H_2_O_2_ system. The characteristics peaks, structure, and morphology were analyzed using Fourier-transform infrared spectroscopy (FT-IR), X-ray diffraction (XRD), and scanning electron microscopy (SEM). g-C_3_N_4_ loaded with 6% Ag exhibited superior photocatalytic performance; the photocatalytic fraction of the degraded materials of the MO and MB solutions reached 100% within 70 and 80 min, respectively, upon adding 1 mL and 2 mL of H_2_O_2_. ·OH and ·O_2_^−^ were the primary active free radicals in the dye degradation process within the synergistic system. Stability tests also demonstrated that the photocatalyst maintained good reusability under the synergistic system.

## 1. Introduction

In recent years, rapid industrialization and population growth have led to an ever-increasing exploitation of natural resources, causing environmental pollution to become a progressively escalating global challenge [1,2,3], where water pollution is particularly serious [4,5,6]. Industrial wastewater contains various toxic substances, among which dyes are particularly hazardous toxic wastes. They are characterized by high acidity, intense coloration, and strong toxicity, potentially causing severe harm to water resources and human health [7]. Consequently, it has become a research focus for scholars to render wastewater containing azo dyes harmless, removing the dye components for discharge into natural water bodies or for reuse.

Photocatalysis, as an advanced oxidation technology, utilizes the radicals generated during the photoreaction process to oxidize pollutants into harmless small molecules such as H_2_O and CO_2_ [8,9]. Researchers like Kun [10] have used photocatalytic technology to degrade typical atmospheric pollutants, with the fraction of removed materials exceeding 80%. Zeng et al. [11] applied photocatalytic techniques to treat low-concentration ammonia nitrogen wastewater, achieving a fraction of removed materials of above 90%. In 1967, Akira Fujishima and his mentor Kenichi Honda discovered that light irradiation pf titanium dioxide (TiO_2_) could split water into hydrogen and oxygen, but the efficiency was not high, so the discovery was temporarily shelved. It was not until 1972 that Fujishima Akira and Honda discovered that ultraviolet light irradiation of a single-crystal titanium dioxide semiconductor electrode could facilitate the photolytic reaction of water, producing hydrogen and oxygen, thus sparking a surge of research into TiO_2_ photocatalytic materials for degrading pollutants. In 2009, graphitic carbon nitride (g-C_3_N_4_), an organic conjugated semiconductor material, was first used as a novel photocatalyst for the production of hydrogen from water [12]. Recently, g-C_3_N_4_, which has a two-dimensional layered structure, narrow bandgap (about 2.7 eV), high chemical stability, nontoxicity, and low production costs, has been widely applied in photocatalytic technology [13,14]. Researchers like Saeed [15] utilized photocatalysis as an effective tool for dye degradation, while Zhou et al. [16] produced photocatalytic concrete based on g-C_3_N_4_, degrading 80% of the methylene blue within 30 min; Pandey et al. [17] achieved 96% and 93% removal of P-nitrophenol and rhodamine dyes, respectively. Li et al. [18], using simulated sunlight, prepared a CeTiO_4_/g-C_3_N_4_ composite material that degraded 95.7% of rhodamine B within 140 min.

Ahad et al. [19] prepared zinc oxide materials that, in conjunction with a H_2_O_2_ system under UV light, decomposed methylene blue solution. After 120 min of irradiation, the photocatalytic efficiency of the multipod and tetrapod ZnO nanostructures increased by approximately 97% and 94%, respectively. Carmen et al. [20] enhanced the photocatalytic performance of SmFe_0.7_Co_0.3_O_3_ thin films through Sr doping and synergistic action with H_2_O_2_, achieving a 100% fraction of degraded material within 120 min. Gong et al. [21] improved the photocatalytic degradation of carbamazepine using FeS2/Fe2O3/organic acids with in situ generated H_2_O_2_. Ryma et al. [22] used fluid dynamics cavitation and TiO_2_-coated glass fibers for the photocatalytic degradation of methyl orange in conjunction with H_2_O_2_. Pan et al. [23] prepared MIL-Fe(53)/modified g-C_3_N_4_ photocatalysts with H_2_O_2_ for the degradation of tetracycline, finding that the 3% Fe–MOF/CM–H_2_O_2_ system could degrade 100% of the tetracycline (10 ppm) within 60 min, 3.6 times the fraction degraded using pure g-C_3_N_4_. Wang et al. [24] conducted the photocatalytic degradation of organic dyes and plant hormones with a Cu(II) composite powder catalyst in conjunction with H_2_O_2_, achieving a high fraction of degraded material of 84.9% for MB under UV light after 90 min, while the degradation efficiency of indoleacetic acid (IAA) exceeded 60.4% after 4 h of irradiation. Yang et al. [25,26] have used graphitic carbon nitride in conjunction with H_2_O_2_ for the photocatalytic degradation of methyl orange and rhodamine B, enhancing the fraction of degraded materials by 2.5 and 3.5 times, respectively, achieving excellent results.

This study synthesized g-C_3_N_4_ through thermal polymerization using urea, followed by the preparation of Ag/g–C_3_N_4_ composite material using AgNO_3_ and g-C_3_N_4_ as precursors through an in situ photoreduction method. Further studies on the morphology, structure, and optical absorption properties of the Ag/g–C_3_N_4_ composite photocatalyst were conducted using XRD, FT-IR, SEM, PL, UV–Vis/DRS, XPS, and other characterization methods. The effects of light source, H_2_O_2_ dosage, initial concentration of target pollutants, dosage of composite material, and solution pH on the photocatalytic degradation of the MO and MB solutions were examined. The stability and reusability of the composite material were tested, and the mechanism of photocatalysis was discussed.

## 2. Results and Discussion

### 2.1. Structural Characterization

#### 2.1.1. XRD Analysis

As shown in Figure 1, XRD was used to determine the crystal structures of CN (g-C_3_N_4_) and Ag-*X*/CN. The characteristic peaks for sample CN at 13.10° and 27.55°, corresponding to the (100) and (002) crystal planes, respectively, can be observed in the diagram. The first diffraction peak is due to the stacking of triazine structure layers within the plane, while the second peak results from the stacking of layers in the conjugated aromatic system, which is a distinctive feature of the graphite structure, indicating the successful preparation of CN. The XRD spectrum of Ag-X/CN is similar to that of CN, suggesting that the loading of Ag did not alter the original growth environment or the existing structure of CN.

#### 2.1.2. Fourier-Transform Infrared Spectroscopy (FT-IR) Analysis

Figure 2 presents the FT-IR spectra of CN and Ag-*X*/CN, showing the functional surface groups of the materials prepared with different composite ratios. The FT-IR spectra of the composites containing varying amounts of Ag are identical to that of pure CN, indicating the successful loading of Ag on CN. The absorption peak at 817 cm^−1^ is attributed to the deformation vibration of the N-C=N bond in the triazine ring. Additionally, the absorption peaks between 1235 and 1660 cm^−1^ are due to the stretching vibrations of C-N bonds, while the broad peak ranging from 3000 to 3500 cm^−1^ is associated with the stretching vibrations of -OH groups on the surface of CN [27].

#### 2.1.3. Scanning Electron Microscopy (SEM)

Figure 3 shows the SEM images of CN at different magnifications, revealing its irregular porous tubular morphology, which resembles the fluffy porous structure of coral. This morphology increases the specific surface area of CN, providing more active sites for degrading target pollutants and thus potentially enhancing the photocatalytic fraction of degraded material [28].

Figure 4 presents the SEM images of Ag-6/CN at different magnifications, showing a thin and dense morphology similar to sheet-like clouds. Compared to CN, Ag-6/CN has a larger specific surface area, allowing more extensive contact with pollutants and providing more active sites, demonstrating a photocatalytic advantage.

#### 2.1.4. Ultraviolet–Visible Diffuse Reflectance Analysis (UV–Vis/DRS)

The optical absorption properties are also a factor affecting the photocatalytic activity of photocatalysts. Therefore, the ultraviolet–visible diffuse reflectance spectra of CN and Ag-6/CN were plotted, as shown in Figure 5a. In this figure, the maximum absorption wavelengths for CN and Ag-6/CN are 448 nm and 457 nm, respectively. Compared to CN, Ag-6/CN exhibits a redshift, indicating that CN has weaker absorption capability for visible light. The loading of 6% Ag on CN broadened the visible light absorption range, enhancing its capacity to absorb visible light and, consequently, improving the photocatalytic performance. The bandgap widths obtained through the Tauc plot method are shown in Figure 5b. The bandgap widths for CN and Ag-6/CN are 2.78 eV and 2.73 eV, respectively. Compared to that of CN, the bandgap width of Ag-6/CN is reduced, facilitating the transfer of photogenerated electrons, which is beneficial for enhancing photocatalytic performance.

#### 2.1.5. Photoluminescence (PL) Spectroscopy Analysis 

Photoluminescence spectroscopy is used to characterize the recombination fraction of photogenerated electron–hole pairs. Lower fluorescence intensity is indicative of more effective separation of photogenerated electrons and holes. To explore the fraction of recombined electron–hole pairs, PL analysis was performed on CN and Ag-6/CN, and the PL spectra were obtained, as shown in Figure 6. Under excitation at 321 nm, the samples exhibited a strong emission peak at approximately 465 nm. The fluorescence intensity of Ag-6/CN was significantly lower than that of CN, indicating that the loading of Ag on the composite material notably suppressed the recombination of photogenerated electron–hole pairs, which was beneficial for reducing the fraction of the recombination carrier.

### 2.2. Study on Photocatalytic Effects of Composite Materials

#### 2.2.1. Effect of Ag-X/CN Composite Ratio on Photocatalysis

The reaction conditions regulated in this study were as follows: at room temperature, the laboratory’s visible light source was a 300 W xenon lamp, the dark adsorption time was 30 min, the light irradiation time was 2 h, the dosage of the photocatalyst was 50 mg, the photocatalyst was Ag-X/CN, the concentration of the target degradant was 20 mg/L (100 mL), and the pH was 7. Among them, the loading ratio of Ag in the catalyst was an experimental variable, which was CN, Ag-2/CN, Ag-4/CN, Ag-6/CN, and Ag-8/CN, respectively. Notably, the dark reaction involved the adsorption of a part of the dye, and each independent experiment was repeated three times.

As shown in Figure 7a, the MO degradation effect by the Ag-*X*/CN composites at different ratios increased in the following order: CN, Ag-2/CN, Ag-4/CN, Ag-6/CN, and Ag-8/CN, with the fraction of degraded materials increasing correspondingly from 31.6%, 49.8%, 52.3%, 57.6%, to 58.0%. However, the fraction of the degraded materials of Ag-6/CN, and Ag-8/CN differed by only 0.4%, and the content of loaded Ag differed by 2%. Since Ag belongs to the precious metals, Ag-6/CN was thus selected as the optimal composite material for degrading MO.

As shown in Figure 7c, the MB degradation effect by the Ag-X/CN composites at different ratios increased in the following order: CN, Ag-2/CN, Ag-4/CN, Ag-8/CN, and Ag-6/CN, with fraction of degraded materials increasing from 65.03%, 71.7%, 80.3%, 81.7%, to 81.8%. Therefore, Ag-6/CN was chosen as the best composite material for degrading MB. It was evident that both excessively low and high amounts of Ag loading reduced the photocatalyst’s degradation effect. Too low Ag loading may not support much CN, while too high Ag loading could cover some active sites on the CN surface. An appropriate amount of Ag loading can improve the separation efficiency of electrons and holes [29,30,31].

As shown in Figure 7b,d and Table 1, pseudo-first-order reaction kinetics were used to fit the MO and MB degradation processes by different Ag-X/CN composites. The fitting effect and fraction of MB degraded material were better compared to those MO, possibly due to the influence of the adsorption during the reaction process. According to the k values and R^2^ listed in Table 1, Ag-6/CN had the highest reaction rate constants during the photocatalytic degradation of MO and MB, which were 0.44 × 10^−2^ cm^−1^ and 1.14 × 10^−2^ cm^−1^, respectively. From Figure 7a,b, it can be observed that Ag-6/CN was more effective in degrading MB than MO.

#### 2.2.2. The Impact of Catalyst and Illumination on Photocatalysis

The reaction conditions regulated in this study were as follows: at room temperature, the visible light source in the laboratory was a 300 W xenon lamp, the dark adsorption time was 30 min, the open light irradiation times of the MO and MB solutions were 70 min and 80 min, the dosage of photocatalyst was 50 mg, the photocatalyst was Ag-6/CN, the synergistic system was H_2_O_2_, the target degradation product concentration was 20 mg/L (100 mL), and the pH was 7.

As shown in Figure 8a, under illumination conditions, the fraction of the Ag-6/CN degraded material of the MO solution was 57.6%. With the addition of H_2_O_2_, the fraction of the degraded material rose to 100%. The introduction of H_2_O_2_ played a significant role in the photocatalytic degradation of the MO solution, facilitating redox reactions between the MO solution and active free radicals, resulting in the degradation of dye wastewater. In the dark reactions, both the Ag-6/CN and H_2_O_2_ exhibited good adsorption properties for the MO solution, but almost no degradation occurred after reaching adsorption–desorption equilibrium. In the other systems, the MO solution was almost undegraded.

As depicted in Figure 8b, with Ag-6/CN and visible light, the fraction of the material of the MB solution degraded by Ag-6/CN after 70 min of illumination was 81.80%. After the dark reaction and the addition of H_2_O_2_, the fraction of material degraded in the MB solution increased to 99.7%. In the absence of a catalyst, with only H_2_O_2_ and light, the MB solution degraded to some extent, with a degraded material fraction of 56.9%. This may have been due to the production of ·OH under visible light irradiation by H_2_O_2_, which further promoted the degradation of the MB solution.

#### 2.2.3. The Impact of H_2_O_2_ Dosage on Photocatalysis

The reaction conditions regulated in this study were as follows: at room temperature, the visible light source in the laboratory was a 300 W xenon lamp, and the dark adsorption time was 30 min. After the dark reaction, different volumes of H_2_O_2_ were added. The open light irradiation times of the MO and MB solutions were 70 min and 80 min, respectively. The dosage of the photocatalyst was 50 mg, the photocatalyst was Ag-6/CN, the target pollutant concentration was 20 mg/L (100 mL), and the pH was 7.

As shown in Figure 9a, with the addition of 0 mL, 0.05 mL, 0.1 mL, 0.5 mL, and 1 mL of H_2_O_2_, the fractions of degraded materials for the MO solution were 49.5%, 79.4%, 85.3%, 99.4%, and 100%, respectively. It was evident that the fraction of the degraded MO material increased progressively with the increase in H_2_O_2_ volume. Particularly, 1 mL of H_2_O_2_ enabled the complete degradation of the MO solution within 70 min of illumination, representing an increase of 50.5% in the fraction of degraded material compared to the scenario without H_2_O_2_.

As depicted in Figure 9b, with the addition of 0 mL, 0.05 mL, 0.1 mL, 0.5 mL, 1 mL, and 2 mL of H_2_O_2_, the fraction of degraded materials in the MB solution were 68.9%, 74.6%, 81.1%, 84.9%, 95.4%, and 99.7%, respectively. Clearly, the fraction of degraded MB material also gradually increased with the increase in H_2_O_2_ volume. Notably, compared to the condition without H_2_O_2_, 2 mL of H_2_O_2_ under 80 min of illumination enhanced the fraction of degraded MB material by 30.7%. Compared to the conditions for the MO solution, the degradation process for the MB solution involved an additional 1 mL of H_2_O_2_ and 10 more minutes of light exposure, yet the fraction of the degraded material was 0.3% lower, indicating that the synergistic H_2_O_2_ system had a more effective degradation impact on the MO solution.

#### 2.2.4. Impact of Initial Solution Concentration on the Photocatalysis by Composite Material

The reaction conditions regulated in this study were as follows: at room temperature, the visible light source in the laboratory was a 300 W xenon lamp, the dark adsorption time was 30 min, 1 mL and 2 mL H_2_O_2_ were added to the MO and MB solutions after the dark reaction, the open light irradiation times of the MO and MB solutions were 70 min and 80 min, the dosage of the photocatalyst was 50 mg, the photocatalyst was Ag-6/CN, the volume of the target degradation products was 100 mL, and the pH was 7. Among them, different initial concentrations of the MO and MB solutions were used as experimental variables, which were 20 mg/L, 25 mg/L, 30 mg/L, 35 mg/L, and 40 mg/L.

As indicated in Figure 10, the fraction of degraded materials for the MO and MB solutions ranged from 86.2% to 100% and 62.7% to 99.7%, respectively, from low to high concentration. As the initial concentration of the MO and MB solutions increased, the fraction of the degraded materials gradually decreased. This was likely due to the higher initial concentrations of the target pollutants, which meant more pollutant molecules, and to the limited capacity of the Ag-6/CN photocatalyst to process these pollutants, thereby affecting the photocatalytic performance and resulting in a smaller fraction of degraded materials.

#### 2.2.5. Impact of Composite Material Dosage on Photocatalysis

The reaction conditions regulated in this study were as follows: at room temperature, the visible light source in the laboratory was a 300 W xenon lamp, the dark adsorption time was 30 min, 1 mL and 2 mL of H_2_O_2_ were added to the MO and MB solutions after the dark reaction, the open light irradiation times of the MO and MB solutions were 70 min and 80 min, the photocatalyst was Ag-6/CN, the target pollutant concentration was 20 mg/L (100 mL), and the pH was 7. Among them, the dosage of the catalyst was the experimental variable, which was 0 mg, 10 mg, 20 mg, 30 mg, 40 mg, 50 mg, or 60 mg.

As shown in Figure 11a, without the addition of the Ag-6/CN photocatalyst, the MO solution was almost undegraded, with a fraction of degraded material of only 3.6%. When the dosages of the Ag-6/CN photocatalyst were 10 mg, 20 mg, 30 mg, 40 mg, and 50 mg, the fraction of degraded materials of the MO solution were 79.1%, 98.3%, 99.6%, 100%, and 100%, respectively. Both 40 mg and 50 mg of the Ag-6/CN photocatalyst achieved the complete degradation of the MO solution within 70 min of light exposure, but from an economic standpoint, the addition of 40 mg of Ag-6/CN photocatalyst is preferred.

As depicted in Figure 11b, with catalyst dosages of 0 mg, 10 mg, 20 mg, 30 mg, 40 mg, 50 mg, and 60 mg, the fraction of degraded materials of the MB solution were 56.9%, 86.1%, 90. 8%, 94.5%, 99.7%, and 98.9%, respectively. As the dosage of the catalyst increased, the fraction of the degraded material of the MB solution also increased, but when the catalyst dosage reached 60 mg, the fraction of degraded material slightly decreased. This decrease may have been due to the excessive amount of catalyst affecting the active sites on the catalyst surface or worsening the light transmittance of the MB solution, thereby impacting the photocatalytic effect. Therefore, the optimal catalyst dosage for the MB solution was 50 mg.

#### 2.2.6. Impact of Solution pH on the Photocatalysis of Composite Material

As shown in Figure 12a, when the initial pH values of the MO solution were 3, 5, 7, 9, and 11, the fractions of degraded materials were 100%, 99.9%, 100%, 100%, and 99.2%, respectively. During the first 40 min of light exposure, the degradation of the MO solution at pH 11 was notably effective but slowed down subsequently. This could have been due to a small amount of alkali promoting the reaction, while an excess of alkali inhibited the redox reaction. Although the MO solutions at pH values of 3, 7, and 9 were completely degraded, the degradation of the material at pH 3 was somewhat faster. This indicated that under weakly acidic conditions, the photocatalytic performance was slightly worse within 70 min of light exposure, possibly because a small amount of acid inhibited the reaction.

As indicated in Figure 12b, when the initial pH values of the MB solution were 3, 5, 7, 9, and 11, the fractions of degraded materials were 76.2%, 89.2%, 99.7%, 99.9%, and 100%, respectively. As the pH value increased, the fraction of the degraded material of the MB solution also increased gradually. This suggests that alkaline conditions are favorable for the migration of photogenerated carriers at the Ag-6/CN interface, while acidic conditions reduce the photocatalytic activity of Ag-6/CN, impacting the photocatalytic effect.

### 2.3. Mechanism of Photocatalytic Degradation of MO and MB by Ag-6/CN with H_2_O_2_ System

To explore the possible photocatalyst mechanisms in under the H_2_O_2_ system in degrading MO and MB, diphenylamine (DPA, 0.5 mM), ethylenediaminetetraacetic acid (EDTA, 0.5 mM), and p-benzoquinone (BQ, 0.5 mM) were used as scavengers for ·OH, h^+^, and other free radicals, respectively. These scavengers were added to the target pollutants containing the photocatalyst in the radical-capturing experiments to analyze the active free radicals playing a major role in the photocatalytic degradation of MO and MB, as shown in Figure 13.

As shown in Figure 13a, after the addition of scavengers, the fraction of degraded MO material in the blank group remained at 100%. After adding EDTA, DPA, and BQ scavengers, the fractions of degraded MO materials were 100%, 59.6%, and 42.0%, respectively, with fractions of inhibition materials of 0%, 40.4%, and 58.0%, respectively. These data indicated that EDTA produced almost no inhibition in the photocatalytic degradation of MO under H_2_O_2_. Both DPA and BQ exhibited certain inhibitory effects on the photocatalytic degradation of MO, with BQ showing a more significant inhibition, 17.6% higher than that of DPA. This suggests that ·OH and ·O_2_^−^ play a role in the photocatalytic degradation process, with ·O_2_^−^ being the most critical active species in the Ag-6/CN in the H_2_O_2_ system for degrading MO solutions.

As shown in Figure 13b, after the addition of scavengers, the fraction of degraded MB material in the blank group was 100%. After adding EDTA, DPA, and BQ scavengers, the fractions of degraded MB materials were 68.5%, 55.2%, and 72.6%, respectively, with fractions of inhibition materials of 31.5%, 44.8%, and 27.4%, respectively. These data indicated that EDTA, DPA, and BQ all exhibited certain inhibitory effects on the photocatalytic degradation of MB. Among them, BQ produced the least inhibition, reducing the fraction of degraded material by 27.4% compared to that of the blank group; EDTA produced slightly more inhibition, only 4.1% more than BQ; DPA produced the most significant inhibition, reducing the fraction of degraded material by 44.8%, about half that compared to the blank group. This suggests that ·OH, h^+^, and ·O_2_^−^ contribute to the removal of pollutants in the photocatalytic degradation process, with ·OH being the most critical active species in the Ag-6/CN with H_2_O_2_ system for degrading MB solutions.

Photocatalysis refers to the chemical reactions induced by photocatalytic materials under the influence of light, representing an intersection of photochemistry and catalytic sciences. Semiconductor photocatalytic materials possess a conduction band structure (CB) and a valence band structure (VB). When semiconductor photocatalytic materials are irradiated by a light source, the electrons from the VB are excited to the CB only if the photon energy is sufficient to be absorbed by the bandgap. Simultaneously, holes (h^+^) are generated on the VB, forming electron–hole pairs (e^−^-h^+^), as shown in Equation (1). The h^+^ and e^−^ can independently undergo oxidation and reduction reactions with the substances adsorbed on the surface of the photocatalytic material. Concurrently, the e^−^–h^+^ pairs that do not participate in the redox reactions may recombine directly on the surface of or inside the catalyst. During the photocatalytic process, H_2_O reacts with holes to produce ·OH and O_2_, as depicted in Equations (2) and (3). O_2_ reacts with electrons to reduce ·O_2_^−^, as shown in equation (4). The addition of H_2_O_2_ triggers the following reaction, where H_2_O_2_ reacts with e^−^ to produce ·OH, as per equation (5). The generated active free radicals (·OH, ·O_2_^−^) and holes degrade pollutants, as indicated in Equation (6) [32,33,34].
(1)$$\begin{array}{c} Semiconductor+hv\to Semiconductor\left(e^{-}+h^{+}\right) \end{array}$$
(2)$$\begin{array}{c} H_{2}O+h^{+}\to \cdot OH+H^{+} \end{array}$$
(3)$$\begin{array}{c} 2H_{2}O+4h^{+}\to {4H}^{+}+O_{2}↑ \end{array}$$
(4)$$\begin{array}{c} O_{2}+e^{-}\to \cdot O_{2}^{-} \end{array}$$
(5)$$\begin{array}{c} H_{2}O_{2}+e^{-}\to \cdot OH+OH^{-} \end{array}$$
(6)$$\begin{array}{c} h^{+}/\cdot O_{2}^{-}/\cdot OH+MO/MB\to degradation\;products \end{array}$$

As shown in Figure 14, based on previous experimental tests and theoretical analysis, we hypothesize the mechanism through which the target pollutants are photocatalytically degraded by the Ag-6/CN in conjunction with the H_2_O_2_ and Na_2_S_2_O_8_ systems under visible light irradiation. The electrons on the VB of Ag-6/CN are excited to the CB, generating photogenerated electrons and holes, e- and h+. The presence of the oxidants Na_2_S_2_O_8_ and H_2_O_2_ reduces the recombination fractions of e^−^ and h^+^, allowing more photogenerated carriers to participate in the degradation of MO and MB. h^+^, ·O_2_^−^, ·OH, and ·SO_4_− can all degrade solutions of methyl orange and methylene blue. However, in the synergistic system, the primary active free radicals degrading the pollutants are ·OH and ·O_2_^−^. O_2_ and H_2_O_2_ undergo reduction reactions to produce ·OH and ·O_2_^−^; H_2_O undergoes oxidation reactions to produce ·OH, which then degrades azo dyes (MO and MB) along with “·O_2_^−”^ into small molecular substances.

### 2.4. Stability Test of Composite Material

Figure 15 shows the degradation cycle graph for the MO and MB solutions under H_2_O_2_ conditions in conjunction with Ag-6/CN. Repeated experiments were conducted under the same reaction conditions. As indicated by Figure 15, after five cycles, the fraction of material of the MO solution degraded by Ag-6/CN decreased from 100% to 89.1%, and for the MB solution, it decreased from 100% to 82.9%. The fraction of degraded materials for both MO and MB solutions remained above 80%, demonstrating that Ag-6/CN exhibited good photocatalytic stability in the degradation process of the MO and MB solutions. The photocatalytic effects were evaluated by recording the concentration changes in the MO and MB solutions before and after the reaction, with the fraction of degraded material calculated according to Equation (7).
(7)$$\begin{array}{c} \eta =\frac{C_{0}-C}{C_{0}}\text{×}100\% \end{array}$$
where η is the fraction of the degraded material, in %; $C_{0}$ is the initial mass concentration of the solution, in mg/L; C is the mass concentration of the solution at time t, in mg/L.

## 3. Experimental Section

### 3.1. Experimental Reagents

The reagents used in this experiment are shown in Table 2. All water used in the experiment was deionized water, and the reagents used were all analytical grade.

### 3.2. Experimental Apparatus

The experimental apparatuses used in this study are shown in Table 3.

### 3.3. Catalyst Preparation

g-C_3_N_4_: A specified amount of urea was placed in a crucible, wrapped in tin foil, and heated in a muffle furnace at a rate of 13 °C/min to 520 °C, and then maintained at this temperature for 2 h. Then, the temperature was increased at the same rate to 550 °C and maintained for another 2 h. After cooling to room temperature, the crucible was removed, revealing a pale yellow solid. This was then ground in a mortar to obtain g-C_3_N_4_, denoted as CN.

Ag/g-C_3_N_4_: First, a certain mass of AgNO_3_ was weighed and dissolved in a 100 mL volumetric flask to make a 4 mM solution of AgNO_3_. Then, 10 mL, 20 mL, 30 mL, or 40 mL of the AgNO_3_ solution was mixed with 0.2 g of g-C_3_N_4_ in a 50 mL beaker. The resulting pale yellow milky solution was sonicated in an ultrasonic cleaner for 30 min to uniformly disperse the g-C_3_N_4_ in the AgNO_3_ solution. Finally, the homogeneously dispersed composite solution was irradiated under a visible light source for 2 h, followed by centrifugation, filtration, and drying to obtain the final products containing 2%, 4%, 6%, and 8% Ag in g-C_3_N_4_ (Ag/g-C_3_N_4_, abbreviated as Ag-*X*/CN), denoted as Ag-2/CN, Ag-4/CN, Ag-6/CN, and Ag-8/CN, respectively.

### 3.4. Structural Characterization Methods

In this experiment, the crystalline structure of the photocatalysts was analyzed using X-ray diffraction (XRD). The X-ray source was a Cu-Kα target (λ = 0.154056 nm). The scan range was 2*θ* = 5–90°, with a scan speed of 2°/min, step size of 0.019°, step duration of 0.22 s, current of 40 mA, and voltage of 40 kV.

Fourier-transform infrared spectroscopy (FT-IR) is a commonly used method for analyzing the functional groups on the surfaces of prepared photocatalytic materials. The prepared materials and pure KBr were first placed in a drying oven for 30 min. Then, they were ground in a mortar to a particle size of < 2 µm and scanned in the range of 4000–400 cm^−1^.

This experiment involved the use of scanning electron microscopy (SEM) to analyze the microstructure of the single and composite photocatalytic materials. Initially, the prepared materials were mounted on black conductive adhesive and then gold-sputtered under a vacuum. The acceleration voltage used was 1.0 kV.

In this experiment, a UV–visible diffuse reflectance spectrophotometry (UV–Vis/DRS) was used to test the light absorption range of the solid powder photocatalyst and to plot the UV–visible diffuse reflectance spectra. The scanning range was 200–800 nm. Based on the light absorption range and the Tauc plot method, as shown in Equation (7), the bandgap widths of the g-C_3_N_4_ photocatalyst and its composites could be deduced. The g-C_3_N_4_ used in this paper was a direct bandgap semiconductor; therefore, n was taken as 1/2.
(8)$$\begin{array}{c} {\left(\alpha hv\right)}^{1/n}=A\left(hv-E_{g}\right) \end{array}$$
where α is the absorbance value; h is the Planck constant, equal to 4.1356676969 × 10^−15^ ev; ν is the frequency; A is a constant; *E_g_* is the semiconductor bandgap width.

Photoluminescence (PL) spectroscopy was used in this experiment to obtain the emission spectra of g-C_3_N_4_ and Ag/g-C_3_N_4_ under 365 nm excitation light to study their electronic structures and optical properties, thus revealing the mechanisms of e^−^–h^+^ migration, trapping, and recombination.

### 3.5. Degradation Experiment Method

#### 3.5.1. Preparation of Methyl Orange and Methylene Blue Wastewater

To prepare stock solutions (1 g/L) and working solutions of methyl orange (MO) and methylene blue (MB), firstly, we weighed 1 g each of MO and MB, which we placed into 100 mL beakers. We added a certain amount of deionized water under stirring until completely dissolved. We transferred the solutions into 1000 mL volumetric flasks, which we made up to the mark with deionized water, resulting in 1 g/L solutions of MO and MB (referred to as stock solutions). We stored these solutions in a refrigerator. Before the experiment, we diluted the MO and MB solutions to the required working concentrations.

#### 3.5.2. Activity Test of Photocatalyst

Using a xenon lamp to simulate sunlight conditions, photocatalytic degradation experiments were conducted with MO and MB as the target pollutants in a photocatalytic performance testing device. The maximum absorption wavelengths for MO and MB were 465 nm and 664 nm, respectively. For each experiment, 50 mg of photocatalyst and 100 mL of target degradant (20 mg/L) were weighed, thoroughly mixed, and placed in a reaction container for a dark reaction for 30 min, with stirring provided by a magnetic stirrer inside a light-proof box. Once adsorption–desorption equilibrium was reached, the xenon light source was turned on to initiate the light reaction. During the light reaction, samples of the target pollutants were taken every 10 min and filtered through a 0.45 µm filter head, and the absorbance was measured at specific wavelengths using a UV spectrophotometer. The photocatalytic effects were evaluated by recording the concentration changes of the MO and MB solutions before and after the reaction, with the fraction of degraded material calculated according to Equation (7).

We used quasi-first-order reaction kinetics to simulate the photocatalytic degradation process of MO and MB, and we plotted the relationship between MO and MB solutions and light exposure time t. We studied the photocatalytic degradation kinetic constant k and correlation coefficient R. The first-order reaction kinetics equation is shown in Equation (9).
(9)$$-Ln\left(C/C_{0}\right)=kt$$
where $C_{0}$ is the initial concentration of the target pollutant, mg/L; C is the concentration of the target pollutant at time t, mg/L.

## 4. Conclusions

The results of characterization indicated that CN and Ag-*X*/CN photocatalytic materials were successfully prepared. It was found that the g-C_3_N_4_ synthesized from urea possesses a fluffy porous structure resembling coral, and Ag-6/CN features a cloud-like thin and dense morphology. Compared to CN, Ag-6/CN photocatalytic material has a larger surface area and lower bandgap energy, providing more active sites, enhancing visible light absorption, promoting the separation of the photogenerated electron–hole pairs, and effectively improving the photocatalytic degradation of the target pollutants. Additionally, due to the high Schottky barrier of Ag, e^−^ is transferred to the Ag and finally transferred to the surface of the photocatalyst to participate in the reduction reaction, which increases the specific area as Ag is added. The results showed that g-C_3_N_4_ loaded with 6% Ag demonstrated superior photocatalytic performance.

In a synergistic H_2_O_2_ system, with initial MO and MB concentrations of 20 mg/L (100 mL), initial pH values of 3 and 11, photocatalyst dosages of 40 mg and 50 mg, and 1 mL and 2 mL of H_2_O_2_ added, respectively, Ag-6/CN achieved a 100% photocatalytic fraction of degraded material for both MO and MB solutions within 70 and 80 min, respectively. After five cycles, the fraction of degraded materials of the MO and MB solutions remained above 80%. The radical trapping experiments indicated that the main active species in the photocatalytic degradation of MO and MB by Ag-6/CN under the synergistic H_2_O_2_ system are ·O_2_^−^ and ·OH.

## Figures and Tables

**Figure 1 molecules-29-03871-f001:**
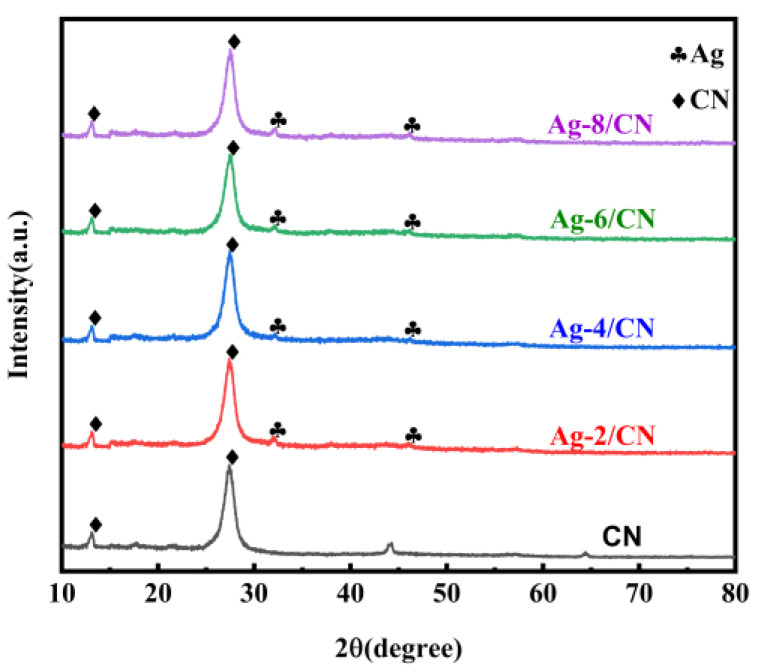
XRD patterns of CN and Ag-X/CN series catalysts.

**Figure 2 molecules-29-03871-f002:**
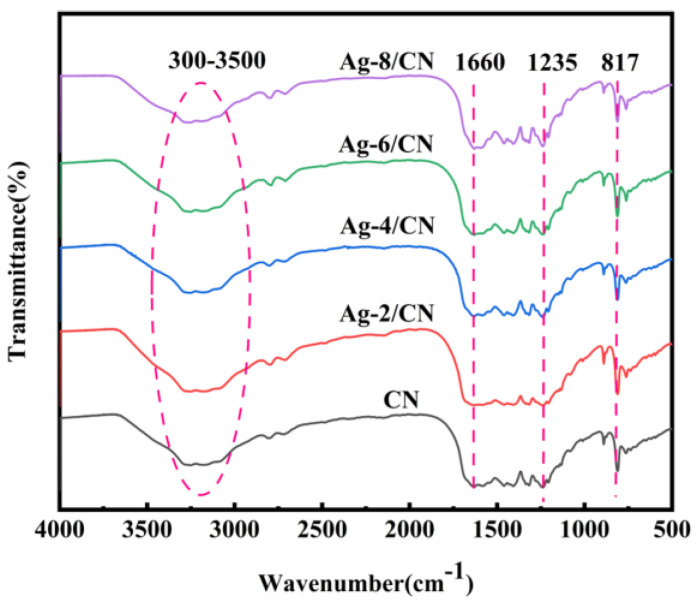
FT−IR spectra of CN and Ag−X/CN.

**Figure 3 molecules-29-03871-f003:**
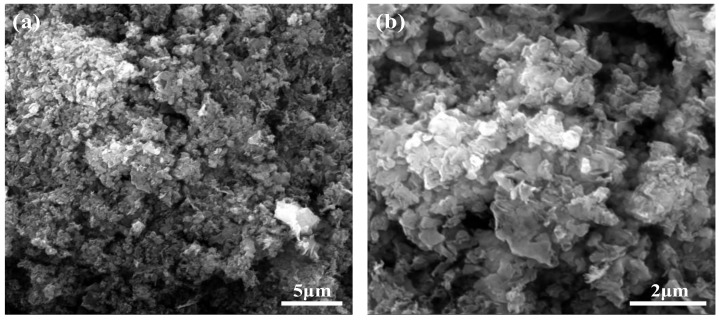
SEM images of CN at different magnifications. (**a**) SEM image of CN at 5 μm; (**b**) SEM image of CN at 2 μm.

**Figure 4 molecules-29-03871-f004:**
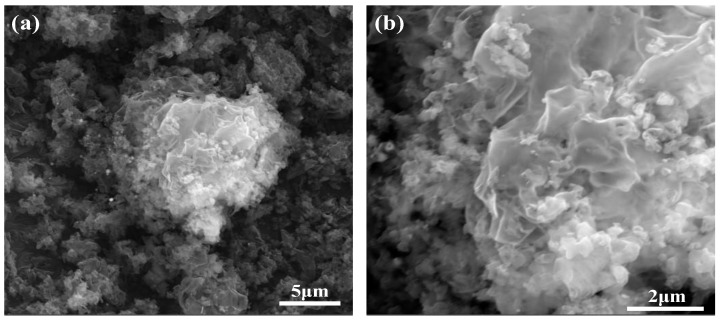
SEM images of Ag-6/CN at different magnifications. (**a**) SEM image of Ag-6/CN at 5 μm; (**b**) SEM image of Ag-6/CN at 2 μm.

**Figure 5 molecules-29-03871-f005:**
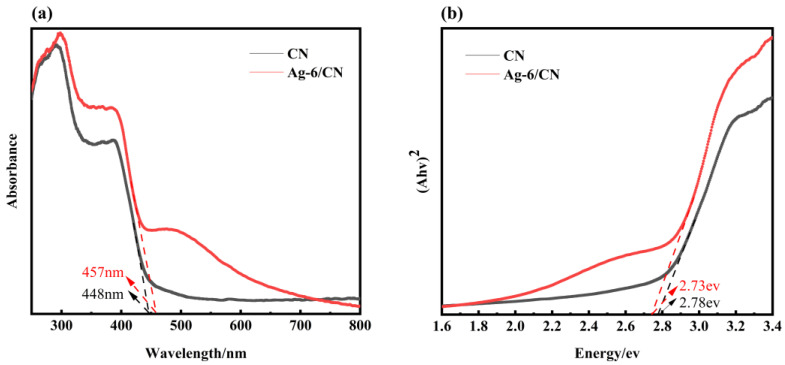
UV–Vis/DRS spectra (**a**) and bandgap width (**b**) of CN and Ag-6/CN.

**Figure 6 molecules-29-03871-f006:**
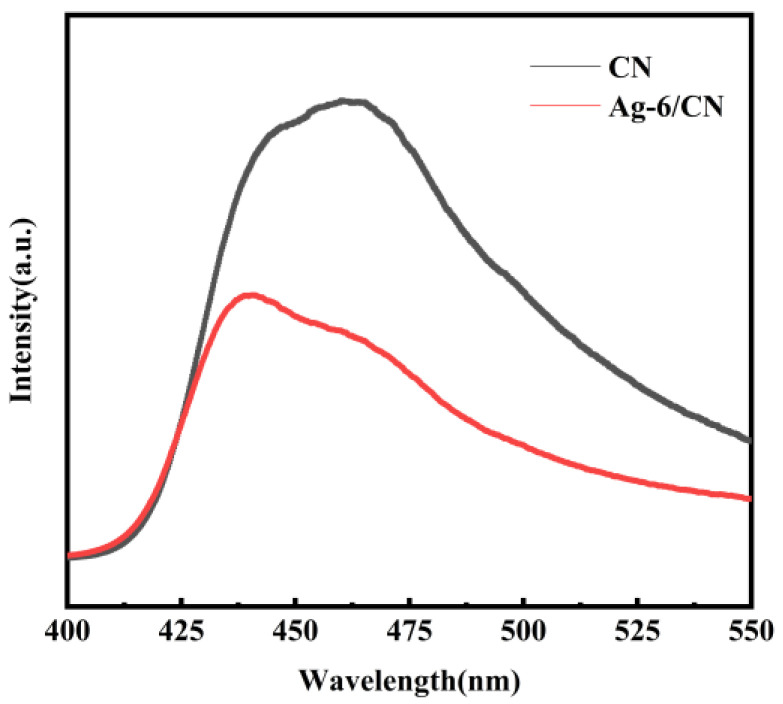
PL spectra of CN and Ag-6/CN.

**Figure 7 molecules-29-03871-f007:**
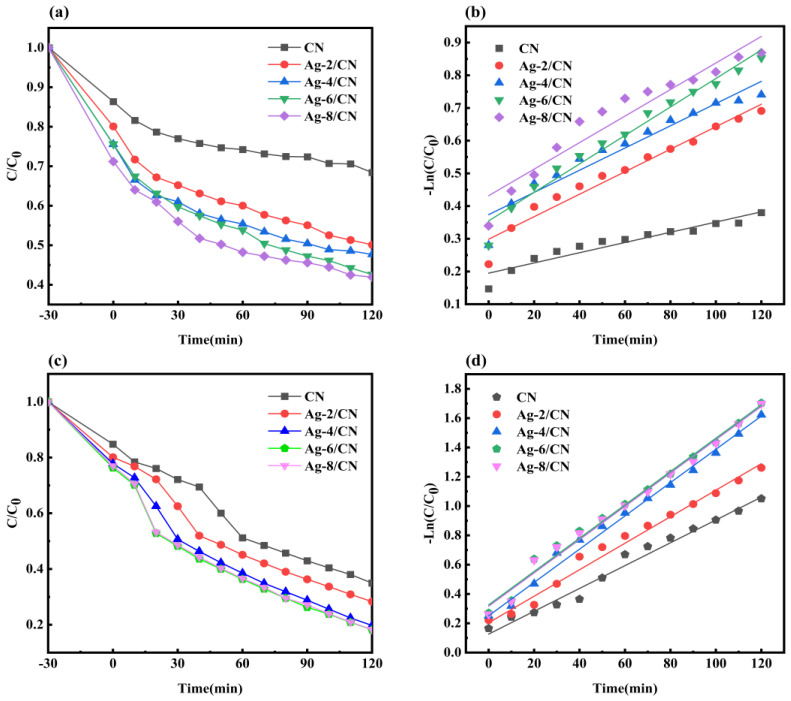
Curve plots of Ag-X/CN degradation of MO and MB with different composite ratios. (**a**) Degradation curve of MO solution; (**b**) quasi-first-order reaction kinetics diagram for degradation of MO solution; (**c**) degradation curve of MB solution; (**d**) quasi-first-order reaction kinetics diagram for degradation of MB solution.

**Figure 8 molecules-29-03871-f008:**
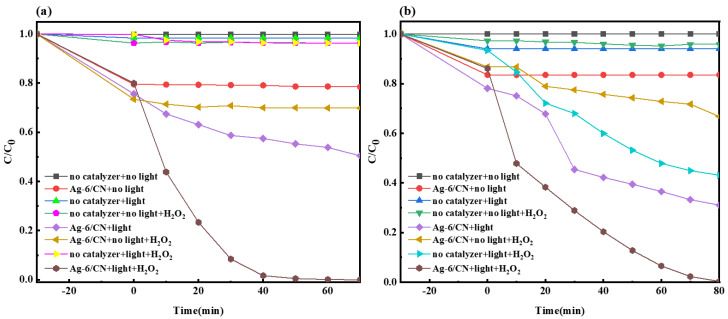
Degradation curves of MO (**a**) and MB (**b**) under different systems.

**Figure 9 molecules-29-03871-f009:**
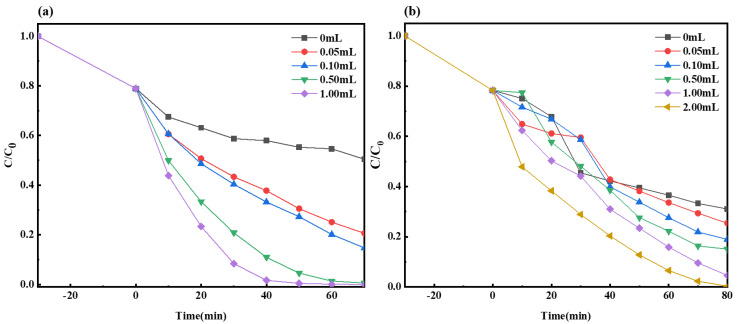
Degradation curves of MO (**a**) and MB (**b**) with different amounts of H_2_O_2_.

**Figure 10 molecules-29-03871-f010:**
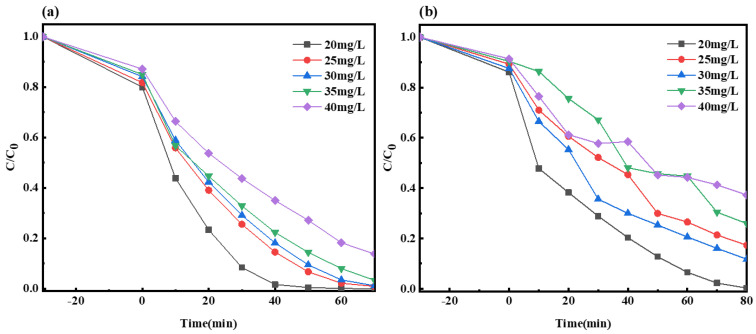
Degradation curves of MO (**a**) and MB (**b**) solutions for different initial concentrations.

**Figure 11 molecules-29-03871-f011:**
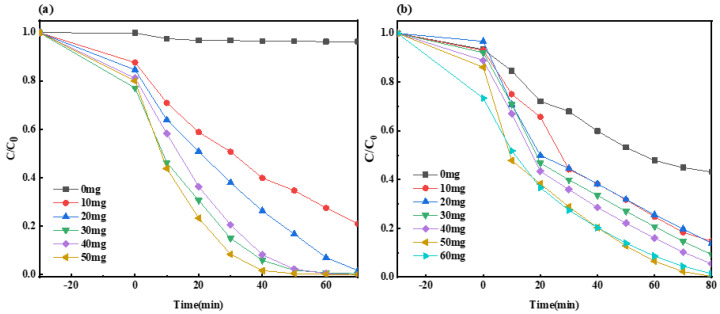
Degradation curves for MO (**a**) and MB (**b**) solutions with varying dosages of catalyst.

**Figure 12 molecules-29-03871-f012:**
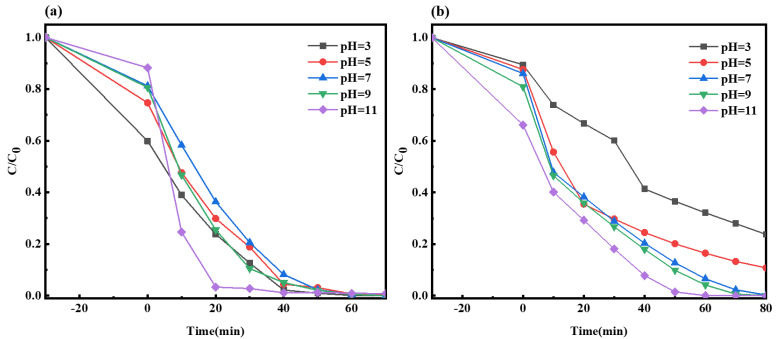
Degradation curves of MO (**a**) and MB (**b**) solutions under different pH.

**Figure 13 molecules-29-03871-f013:**
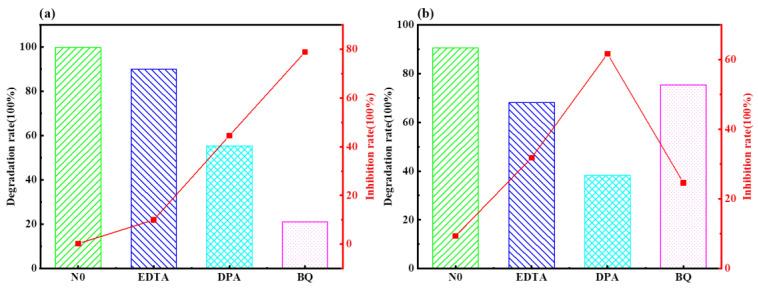
Degradation diagrams of MO and MB using different capture agents. (**a**) H_2_O_2_ degradation of MO; (**b**) H_2_O_2_ degradation of MB.

**Figure 14 molecules-29-03871-f014:**
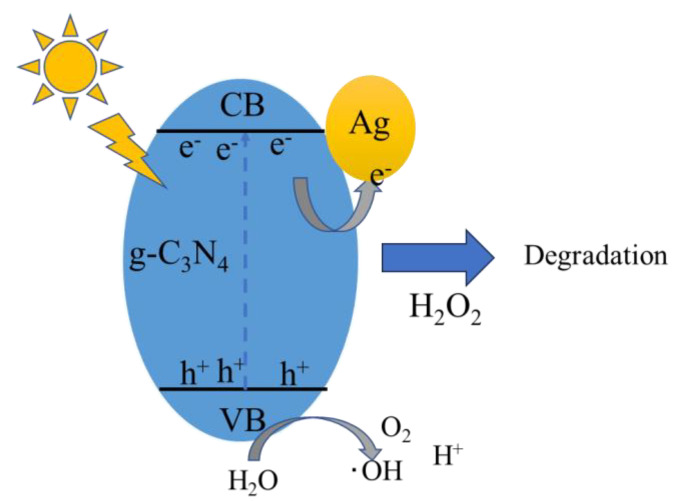
Diagram of the mechanism of the photocatalytic degradation of MO and MB in synergistic Ag−6/CN oxidation system.

**Figure 15 molecules-29-03871-f015:**
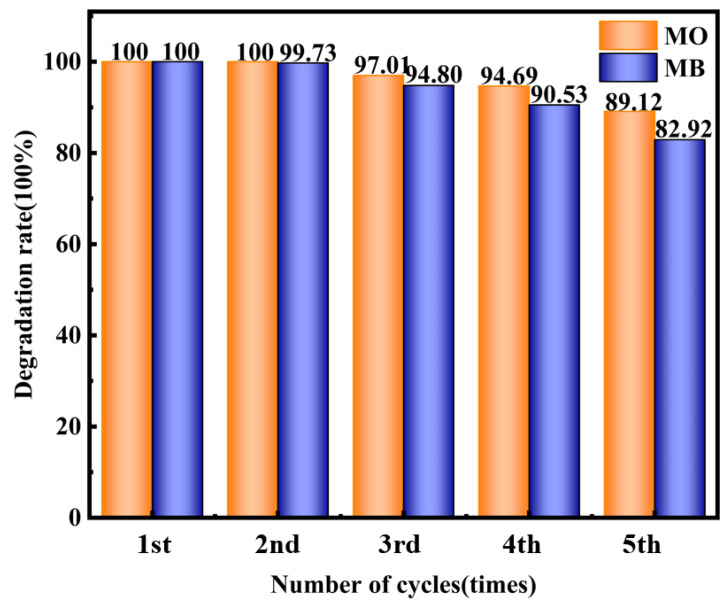
The stability of degradation rate with number of cycles.

**Table 1 molecules-29-03871-t001:** Quasi-first-order reaction kinetics parameters for the degradation of MO and MB by Ag-X/CN with different composite ratios.

	Methyl Orange (MO)		Methylene Blue (MB)	
	K (×10^−2^/min^−1^)	R^2^	K (×10^−2^/min^−1^)	R^2^
CN	0.156	0.902	0.779	0.968
Ag-2/CN	0.344	0.955	0.905	0.978
Ag-4/CN	0.340	0.928	1.136	0.987
Ag-6/CN	0.435	0.969	1.138	0.987
Ag-8/CN	0.406	0.915	1.137	0.987

**Table 2 molecules-29-03871-t002:** The list of experimental reagents.

Name	Chemical Formula	Manufacturer
Urea	CH_4_N_2_O	Tianjin Damao Chemical Reagent Factory, Tianjin, China
Silver Nitrate	AgNO_3_	Sinopharm Chemical Reagent Co., Ltd, Shanghai, China
Anhydrous Ethanol	C_2_H_6_O	Wuxi Zhanwang Chemical Reagent Co., Ltd, Wuxi, China
Methyl Orange	C_14_H_14_N_3_NaO_3_S	Tianjin Guangfu Technology Development Co., Ltd, Tianjin, China
Methylene Blue	C_16_H_18_ClN_3_S·3H_2_O	Shanghai Zhanyun Chemical Co., Ltd, Shanghai, China
Hydrochloric Acid	HCl	Sinopharm Chemical Reagent Co., Ltd, Shanghai, China
Sodium Hydroxide	NaOH	Xilong Scientific Co., Ltd, Shantou, China
p-Benzoquinone	C_6_H_4_O_2_	Shanghai McLean Biochemical Technology Co., Ltd, Shanghai, China
Diphenylamine	C_12_H_11_N	Tianjin Siyou Fine Chemicals Co., Ltd, Tianjin, China
Methanol	CH_3_OH	Tianjin Damao Chemical Reagent Factory, Tianjin, China
EDTA	C_10_H_16_N_2_O_8_	Shanghai McLean Biochemical Technology Co., Ltd, Shanghai, China
30% Hydrogen Peroxide	H_2_O_2_	Sinopharm Chemical Reagent Co., Ltd, Shanghai, China

**Table 3 molecules-29-03871-t003:** The list of experimental apparatuses.

Name	Model	Manufacturer
Magnetic Stirrer	8S-1	Changzhou Guoyu Instrument Manufacturing, Changzhou, China
Electronic Balance	FA2004	Shanghai Shunyu Hengping Scientific Instrument Co., Ltd, Shanghai, China
Electric Constant Temperature Blast Drying Oven	DHG-9101-1SA	Shanghai Sanfa Scientific Instrument Co., Ltd, Shanghai, China
Muffle Furnace	KSL-1100X	Hefei Kejing Material Technology Co., Ltd, Hefei, China
Ultrasonic Cleaner	KQ-100B	Kunshan Jielimei Ultrasonic Instrument Co., Ltd, Kunshan, China
Xenon Lamp Light Source	CEL-HXF300	Beijing Zhongjiaojin Yuan Technology Co., Ltd, Beijing, China
Optical Dark Box	GXAS345	Beijing Newbit Technology Co., Ltd, Beijing, China
Photocatalytic Glass Reactor	KW100	Beijing Newbit Technology Co., Ltd, Beijing, China
UV–Vis Spectrophotometer	UV-26001	Suzhou Shimadzu Instrument Co., Ltd, Suzhou, China
Vacuum Freeze-Dryer	FD-1A-50	Beijing Boyikang Experimental Instrument Co., Ltd, Beijing, China

## Data Availability

Data are contained within the article.

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
