# Peer review of "Synergistic Ag/g–C3N4 H2O2 System for Photocatalytic Degradation of Azo Dyes"

_molecules, 2024, doi:10.3390/molecules29163871_

Round 1

Reviewer 1 Report

Comments and Suggestions for Authors

The manuscript is well written, and the authors presented the results properly.

Before listing some technical issues, the only "problematic" sentence is the one presented in lines 172 and 173, since the lower PL intensity doesn't always indicate a lower recombination rate (in some instances it indicates quite the opposite). The authors should cite appropriate papers to confirm their claim.

Regarding the technical issues:

1. Presented figures seem to be of a lower quality (blurry). I suppose this is a technical issue that can be easily resolved.

2. For results presented in percent, there should be a spacing between the numerical value and the percent sign (not "10%" but rather "10 %").

3. line 57: reference [19] is not noted as the other references (the name of the first author was not used, just "A. A. M.").

4. line 90: Please, fix the spacing issues.

5. In 2.2.1, please add (to the textual part) what the irradiation times were (it is clear from the figure, however, for completeness, this information should be part of the text as well).

6. line 197/198: Please, reformulate this sentence.

7. Figure 15: The description is not complete (it is missing the "(a)").

8. line 360: The "-" is written in more the one manner, therefore I suggest that you chose one and make the manuscript coherent.

9. In 2.4, please refer to the equation you used (I suppose equation 8).

10. The 3.5.1. section should be written in past tense.

Comments on the Quality of English Language

Quality of the English language is satisfying.

Author Response

Thanks for your professional comments, the detailed revisions are presented in the document as below.

Reviewer 2 Report

Comments and Suggestions for Authors

The authors synthesized graphitic carbon nitride via thermal polymerization and depositing noble metal silver onto g-C3N4 through photoreduction. Methyl orange (MO) and methylene blue (MB) were targeted as pollutants in photocatalytic experiments under visible light, in conjunction with an H2O2 system. The authors provided a comprehensive characterization of the prepared photocatalyst and discussed the photocatalytic mechanism, which is a relatively complete study. To improve the quality of the article, I suggest making some modifications, as follows.

1. The resolution of the images is not high, please provide higher resolution images.

2. In the PL spectra, the author tested Ag-6/CN. What about other concentrations?

3. Figure 7. XPS image, image is not appropriate.

4. To deepen the mechanism analysis, it is recommended to test the specific band potential and then determine whether the photo generated electrons and holes have sufficient reducibility and oxidizability to generate superoxide radicals and hydroxyl radicals. The schematic diagram provided now is insufficient.

5. The addition of Ag increased the specific surface area. Please have some necessary discussions and analyze the reasons.

6. What is the specific wavelength of the light source?

7. Papers can be used as references as follows:

https://doi.org/10.1016/j.molstruc.2023.136440

https://doi.org/10.1515/ntrev-2022-0483

https://doi.org/10.1016/j.apcatb.2020.119034

Author Response

(The authors gave the same response as above.)

Reviewer 3 Report

Comments and Suggestions for Authors

The manuscript of Wang et al. reports the utilisation of Ag-loaded graphitic carbon nitride microparticles as a photocatalyst for the degradation of model pollutants, methyl orange and methylene blue. The study revealed a synergistic effect of exposition to light and hydrogen peroxide for the degradation of selected dyes. The topic is actual, findings are promising and have the potential to attract the interest of chemists working in the field of decontamination. Therefore, the work generally deserves publication. However, to my opinion, the interpretation of some experiments is a bit cloudy and some conclusions are very speculative. Therefore, I recommend to publish the manuscript after appropriate alterations will be made.

Major objections:

General – do not use the term “degradation rate”, it is not a rate at all. E.G. “fraction of degraded material” is a more appropriate term. Furthermore, some of the dye may not decompose but just adsorb on the surface of the material.

Please, specify basic experimental conditions also in the main text (reaction volume, concentration of dye, amount of solid material).

The amount of Ag loaded on the CN material is just taken from a presumption that all Ag was reduced and adsorbed on the CN particles. Some elemental analysis should be appropriate.

lines 130-131 – how is the mentioned pore distribution seen in Figure 5b?

Figure 5 – why does the absorbed amount lie below the zero (red curve)? How was the sample pre-treated? Use arrows to indicate experiments with increasing/decreasing relative pressures.

line 145 – reflectance spectrum was measured, so there is no absorbance... Wasn’t it reflectance? Kubelka-Munk function? Correct also axes description in Figure 6. If diffuse reflectance spectra were measured, the direct comparison of “absorption” between different samples is doubtful/principally incorrect (discussion at lines 146-149, description of bandgap calculation at lines 425-435). Please discuss the conditions of measurements and prove that the spectra obtained for different solid samples are comparable.

Section 2.1.7 – similar to above – if PL spectra were taken for the solid samples, their direct comparison is problematic. Please, convince the reader that it can be directly compared by explicitly stating the experimental procedure/conditions.

The degradation rate is not a rate, but just a portion decomposed under very specific conditions and after a specific time. Please, re-formulate and specify conditions also in the main text (chapter 2.2.1)

Figure 9c – obviously, some experimental error occurred in concentration determination between c/c0 0.5-0.7 (drift of baseline, wrong calibration...?). Were the measurements repeated? It leads to wrong numeric results (but a relative effectivity of different materials is right).

Unreasonably precise rate constants and “reaction rates” are present (especially when R^2 is only 0.90-0.96), Table 2 (see also note in the „Formal“ section below).

line 248 – discussion of a difference between 100 and 99.7 % course of the reaction is a bit strange – in both cases, the dye was destroyed. Are you sure that 0.3 % of MB remained? How did the spectrum look like? Or, only absorbance at selected lambda_max was measured? The concentration of the dyes declared to 4 digits is unreasonably accurate, what was the scattering of the data in different independent experiments?

Discussion in lines 255-257 is in the conflict with a legend in charts shown in Figure 12 (from low to high; reverse order?).

lines 285-288 – very speculative, it seems, that the reaction was finished, and a very low “concentration” of MO can be attributed to baseline drift. Was the experiment repeated? How many? Was measured whole spectrum, or only absorbance at one selected wavelength?

Please, explain which scavenger interfere with which reactant (OH radical, O2+ radical, hole+) – text at lines 302-305.

Please, do not use the term “inhibition rate”, it is not a rate (see notice regarding the term “degradation rate”).

What was the role of AE? (line 303)

Please, balance the equations 3 and 4.

Figure 16> reaction on the right-bottom is meaningless – O2 + H+ + h+ will not give OH rad + O2+ rad, please, re-design the Figure.

line 391 – tin foil was melted under these conditions... What about tin contamination of the material? Please, specify in more detail.

Formal:

Some shorts are not defined or are used before their explanation (line 54, PNPO, RhB; line 72, IAA; line 303, DPA, EDTA, BQ, AE). Define holes h+.

Line 121 – surface area is discussed before the information is given. “See below” should be added to refer next chapter 2.1.4.

Presented numbers are often rounded to the unreasonable number of digits – Table 1, specific area to 5 digits, pore volume to 4 digits, and pore diameter to 4 digits; “degradation rate” systematically to 4 digits, etc. Were the measurements repeated? What was the result’s scattering?

Figure 9 – (a) is missing before “Degradation curve of MO...”, line 203

Please, specify explicitly that “dark reaction” is just an adsorption of a part of the dye` explain it in the main text, as a service for the reader (i.e. why absorbance was decreased between time minus 30 min and the start of the experiment).

Author Response

(The authors gave the same response as above.)

Round 2

Reviewer 2 Report

Comments and Suggestions for Authors

After modification, the quality of the article has been improved, and it is recommended to accept publication.

Author Response

Thank you. We really appreciate your professional suggestions.

Reviewer 3 Report

Comments and Suggestions for Authors

The text of manuscript of Wang et al. was significantly improved. However, in my opinion, there still some aspects remain which should be addressed before publication. They are:

General – some terms “removal rate” survive, please, change.

The authors re-measure data shown in former Figure 9c (now in Figure 7c), but there is still some unexpected break. However, it can be acceptable in this form, but the data were used in former Figure 9d, and it was not reflected in new Figure 7d – please, re-calculate/draw it according to the new dataset.

Presented fraction of degradation rounded to 4 digits is meaningless – according to weights of catalyst (2 digits), volume (exactly measured to 2-3 digits), common variation in absorbance measurement, etc., only 2-3 digits are relevant. How many independent experiments were performed? What was scattering? For sure bigger than 4 digits. Please, use e.g. 52 or 52.2 instead of 52.23…

Reaction (4) is not balanced, should be + 1e-

Author Response

Thank you for your comment. Please refer to the document for revision details.
